# Revisiting functioning recovery in persons with spinal cord injury undergoing first rehabilitation: Trajectory and network analysis of a Swiss cohort study

Jsabel Hodel[1]*, Carla Sabariego[1,2,3], Mayra Galvis Aparicio[1☉], Anke Scheel-Sailer[1,2,4☉], Vanessa Seijas[1,2,3☉], Cristina Ehrmann[1]

1 Swiss Paraplegic Research, Nottwil, Lucerne, Switzerland, 2 Faculty of Health Sciences and Medicine, University of Lucerne, Lucerne, Switzerland, 3 Center for Rehabilitation in Global Health Systems, Faculty of Health Sciences and Medicine, University of Lucerne, Lucerne, Switzerland, 4 Swiss Paraplegic Centre, Nottwil, Lucerne, Switzerland

☉ These authors contributed equally to this work.
* jsabel.hodel@paraplegie.ch

**Data Availability Statement:** The datasets generated and analyzed during this study are not publicly available due to the commitment of SwiSCI

## Abstract

Information about an individual's functioning and its longitudinal development is key to informing clinical rehabilitation. However, the description and understanding of the detailed longitudinal course of functioning, i.e., functioning trajectories, is rare in the current SCI literature. The aim of this study was to re-estimate previously identified functioning trajectories of individuals with spinal cord injury (SCI) undergoing initial rehabilitation in Switzerland using trajectory analysis, and to identify highly influential functioning domains that could become trajectory-specific targets for clinical interventions using network analysis. The study was based on data from the Swiss SCI Cohort Study and included individuals with SCI (N = 1099) who completed their rehabilitation in one of four collaborating centers between May 2013 and March 2022. For the trajectory analysis, functioning was operationalized using the total sum score of the Spinal Cord Independence Measure version III (SICM III), which was assessed at up to four time points (T1-T4) during rehabilitation. For the network analysis, individual SCIM III items were used to operationalize relevant functioning problems at T1 (admission) and T4 (discharge). The re-estimation of trajectory analysis confirmed the previously identified mean functioning trajectory classes of *stable high functioning* (N = 239; 21.75%), *early* (N = 33; 3.00%), *moderate* (N = 753; 68.52%), and *slow* (N = 74; 6.73%) *functioning improvement*. The network analysis revealed highly connected functioning problems at T1 for the *moderate functioning improvement class*, including "*Feeding*", "*Dressing upper body*", and "*Dressing lower body*", "*Mobility in bed*", and "*Use of toilet*". These functioning domains might indicate potential trajectory-specific targets for clinical interventions. This study has increased our knowledge about functioning trajectories of individuals with SCI undergoing initial rehabilitation in Switzerland and its findings may inform discussions about the application and use of functioning trajectories in clinical practice. Due to the exploratory nature of this study, further research is needed to confirm the findings presented.

**Funding:** The authors received no specific funding for this work.

**Competing interests:** The authors have declared that no competing interests exist.

## Introduction

Information about an individual's functioning and its longitudinal development is key to informing clinical rehabilitation. Rehabilitation is the health strategy that aims to optimize people's functioning, and accordingly, functioning has been defined as the primary outcome for rehabilitation [1]. According to the WHO's International Classification of Functioning, Disability and Health (ICF) [2], functioning is the outcome between a given health condition of a person and the environment in which this person lives. Consequently, for rehabilitation to be effective, functioning must be continuously monitored, and the provision of interventions must be interactively adapted to a person's functioning status and contextual situation, as described, for example, in the approach of sequential Rehab-Cycles [3–5]. For people with a spinal cord injury (SCI), rehabilitation is a lifelong process that is associated with a high degree of complexity: Not only in terms of the sheer range and variety of impacts that the injury can have on an individual's functioning, whether at the level of body structures and functions, activities, or participation in life situations. But also, in terms of the interaction of functioning with the occurrence of secondary health conditions such as pressure injuries or pulmonia, or with adaptations to a person's environment such as the provision of a wheelchair, and the corresponding changes in functioning status over time.

In the current SCI literature, the description and understanding of the detailed longitudinal course of functioning, i.e., functioning trajectories, using multiple follow-up time points is rare. Recently, unobserved subgroups (i.e., classes) of persons with common overall functioning trajectories during initial SCI rehabilitation have been identified [6] and recovery trajectories of physical function at up to two years after the injury have been modelled [7]. The authors of these studies have stressed the importance of trajectories for clinical practice in terms of providing "recovery roadmaps" for clinicians and patients [7] and in terms of monitoring individuals in relation to a group of similar patients (benchmark) [6]. Nevertheless, a more detailed examination of functioning trajectory classes (hereafter also referred to as functioning trajectories for simplicity) of people with SCI and their clinical relevance is still needed to concretize and refine the proposed suggestions. A prerequisite for their use in clinical practice is a deeper understanding of the identified trajectories, taking into account the complexity of functioning in persons with SCI. This includes examining the trajectory-specific prevalence of relevant functioning problems, such as bladder functions [8, 9], skin functions [10, 11], sensation of pain [12, 13], or emotional functions [14, 15], as well as the detailed association structures among these problems. Such insights have the potential to inform use cases of functioning trajectories in practice and reveal trajectory-specific areas of interest for intervention targets to support tailored rehabilitation.

The objective of this study is to increase the knowledge about functioning trajectories of people with SCI undergoing initial rehabilitation in Switzerland and to identify highly influential functioning domains that could become trajectory-specific targets for clinical interventions. This has been pursued in two ways: First, we aimed to re-estimate the overall functioning trajectories identified by Hodel et al [6] based on an updated dataset, and to extend the analyses in a multivariate fashion to domain-specific functioning outcomes (self-care, respiration and bladder/bowel management, mobility). Second, we aimed to examine the association structures among relevant functioning problems for each trajectory in order to identify trajectory-specific and highly influential functioning problems and potential corresponding targets for interventions.

## Methods

### Study sample

This study reports on the inpatient data from the Inception Cohort of the Swiss SCI Cohort Study (SwiSCI) collected between May 1, 2013 and March 16, 2022 (date of data access). The Inception Cohort includes people with a newly diagnosed SCI who are undergoing initial rehabilitation at one of four specialized clinics in Switzerland. The recruitment process for the SwiSCI Inception Cohort includes different consent scenarios that correspond to the following participation statuses of eligible individuals: (1) No participation (denial of any data collection or use), (2) Minimal data set assessment (MDS; no consent to the SwiSCI Inception Cohort, but general consent to the use of routine clinical data for research purposes), (3) full data set assessment (FDS, written informed consent to the SwiSCI Inception Cohort). For the FDS, up to four data collections during initial rehabilitation are conducted: 4 (T1), 12 (T2), and 24 weeks after SCI diagnosis (T3), as well as at discharge (T4). The MDS includes data assessed at the two time points T1 and T4. Detailed information about the recruitment process and design is published elsewhere [16, 17]. The SwiSCI Inception Cohort has been approved by all responsible regional ethic committees [17].

### Measures

Participant characteristics included basic demographics, namely age at injury (years) and sex (male/female), as well as injury characteristics, including etiology (traumatic/non-traumatic), injury level (paraplegia/tetraplegia) and injury severity (American Spinal Injury Association Impairment Scale (AIS) grades A, B, C, D, E [18]), and length of stay (days). To characterize the study sample, descriptive analyses were conducted.

For the first aim, the primary outcomes were overall and domain-specific functioning, operationalized by the Spinal Cord Independence Measure version III (SCIM III) total and sub scores, respectively. The SCIM III assesses the independence of people with SCI in performing activities of daily living (ADL) related to self-care, respiration and bladder/bowel management, and mobility [19]. SCIM III total scores range from zero to 100, the self-care subscale from zero to 20, and the further subscales each from zero to 40. Higher scores indicate higher independence in ADLs. In the Inception Cohort, the date of a SCIM III assessment is documented in days since SCI diagnosis. Following previous analysis [6], assessment dates were converted to days since admission to initial rehabilitation.

For the second aim, relevant functioning problems were operationalized by variables reflecting the functioning categories of the Brief ICF Core Set for SCI (early post-acute context) [20] or additional statistically relevant ICF categories according to Ballert et al [21]. To be included in the analyses, variables had to be assessed at both timepoints T1 and T4 and have less than 20% missing observations at each time point. Selected variables included 18 SCIM III items (see S1 Table). No suitable variables were identified for six categories of the Brief ICF Core Set (*b152 emotional functions, b280 sensation of pain, b730 muscle power functions, b735 muscle tone functions, b810 protective functions of the skin, d445 hand and arm use*) and for ten of the relevant categories of Ballert et al that are reflected in the SwiSCI Inception Cohort (*b126 temperament and personality functions, b130 energy and drive functions, b270 sensory functions related to temperature and other stimuli, b415 blood vessel functions, b430 hematological system functions, b445 respiratory muscle functions, b710 mobility of joint functions, d240 handling stress and other psychological demands, d770 intimate relationships, d930 Religion and spirituality*). Because of this limited set of variables, and because the SCIM III had an acceptable number of missing observations, it was decided to include the additional SCIM III item

"*Stair management*", which covers the ICF category *d455 moving around*. The main ICF categories covered by the SCIM III items were grouped into a total of 7 ICF domains. A detailed overview of the SCIM III items, study labels and corresponding ICF domains can be found in S2 Table. In addition, patient- (sex, age) as well as injury-related (injury level and severity) covariates were included in all network models.

## Rasch analysis

In preparation for the trajectory analysis, the raw SCIM III total and sub scores were interval-transformed using Rasch analysis and a Partial Credit Model [22, 23]. Rasch analysis of SCIM III scores was performed to ensure that changes in functioning over time are accurately assessed and comparable. Separate analyses were performed for the total and sub scores based on calibration samples. The calibration sample for each analysis included N = 400 observations of available participants from the SwiSCI Inception Cohort or the MDS assessment and was developed based on random sampling with equal distributions of assessment time points (T1-T4) and SCIM III scores without extremes (1–99). Non-consecutive scorings of response categories were converted to consecutive scorings (see S2 Table for an overview on both scoring systems). For each model, fit was assessed by means of overall item and person fit (residuals: SD<1.4 indicating good fit), individual item fit (residuals: non-significant Bonferroni-corrected p-values), and the chi-squared test statistic of the item-trait interaction (non-significant p-value). Reliability of the scores was assessed by the person separation index (PSI>0.7). In addition, the underlying assumptions of the Rasch measurement model were iteratively assessed: Unidimensionality (inspection of paired t-test between person estimates based on two item subsets according to factor loadings on first principal component of the residuals and person estimates based on all items), local independence between items (inspection of the residual correlation matrix), group invariance (inspection of differential item functioning, DIF), and stochastic ordering of item response options (inspection of threshold ordering). If local dependence was present, a so-called testlet approach was applied: Response options of locally dependent items were summed up to create super-items that were no longer locally dependent. Final models of each Rasch analysis were repeated on validation samples (same specifications as for the calibration samples). A solution was discarded if it did not fit well on the validation sample.

As a similar Rasch analysis of the SCIM III total score using SwiSCI data already exists [6, 24], the robustness of the derived interval-based overall functioning scores was investigated by calculating the Pearson correlation coefficient between currently and previously developed scores for each time point of assessment T1-T4 based on a common subsample of participants across studies. The derived overall functioning scores were considered robust if the Pearson correlation coefficient r>0.9.

## Trajectory analysis

For the first aim, latent process mixed models (LPMMs) [25–27] with latent classes were implemented to identify the number of underlying functioning trajectory classes that best described individual's functioning courses during clinical rehabilitation. To model the trajectories, the interval-based SCIM III total and three subscales were used, respectively. Analyses were conducted separately and followed the process described in detail elsewhere [6]: First, the analyses included the estimation of three different LPMMs, each with a different specification for the parameterized link function (linear function, quadratic I-splines functions with two or three knots at the respective quantiles of the SCIM III sum score distribution). These models were compared in terms of the Akaike information criterion (AIC) and the model with the

best-fitting link function was identified by lowest AIC. Second, two different sets of model specifications (set 1: variability of trajectories between individuals is fixed across classes; set 2: variability of trajectories between individuals is allowed to vary across classes) were estimated with increasing number of classes (1–6) and including the best-fitting link function identified in the previous step. Models were compared and the best-fitting LPMM was selected based on the following criteria: Lowest Bayesian information criterion (BIC), sample-size adjusted BIC (SSABIC), and AIC (with preference for BIC over SABIC and AIC [28]), interpretability and convergence, entropy (higher values preferred) and class sample sizes (>5% of the study sample preferred). No covariates were included in the specified models. Alternative model specifications were fitted and are available from the authors on request.

## Network analysis

For the second aim, an exploratory network analysis approach was applied to investigate the class-specific association structures among relevant functioning problems within the identified trajectory classes. To do so, missing data on variables of interest were imputed using the non-parametric method MissForest which is based on random forest [29].

Network analysis is a tool for estimating and exploring comprehensive association structures among variables of interest, such as functioning categories [30, 31]. Once estimated, dependencies between variables of interest can be visualized according to graphical networks. Such networks consist of nodes, representing the variables of interest, and edges and edge weights, representing the estimated associations between the variables and their strengths. The *presence* of an edge between two variables in a network indicates an undirected dependence between them, controlled for all other variables in the network. Similarly, the *absence* of an edge between two variables in a network indicates their conditional independence: These two variables have no dependency relationship, controlled for all other variables in the network. Importantly, network analysis also provides the ability to assess the importance of a variable in a network graph in terms of centrality indices: These indices are a measure of the connectivity of a variable in the network in terms of their edge weights. Under the assumption that addressing a functioning problem that is a highly connected variable will have a greater impact on the network and on overall functioning than in a less connected variable, we considered such variables to be potentially interesting targets for rehabilitation interventions.

For this study, mixed graphical models (MGMs) were used to account for the mixed types of variables (binary, ordinal, continuous) [32]. This approach estimates node-wise regularized general linear regression models and combines the estimated regression coefficients by averaging to represent the edge weights in a network model. Statistical regularization is performed by using the least absolute shrinkage and selection operator (LASSO) to control the number of spurious correlations. The LASSO tuning parameter, which controls the sparsity of a network, was automatically selected from a collection of networks under different values of the tuning parameter by minimizing the Extended BIC (EBIC) [33, 34]. The corresponding EBIC hyper-parameter, which controls the degree to which simpler models are preferred, was set manually based on the exploration of a set of different values (0/0.25/0.5). For this study, ordinal and skewed continuous variables were handled using rank-order transformation [35]. Moreover, the default AND-rule to combine regression estimates was used (both regression coefficients need to be non-zero for an edge to be present). For each trajectory class, a separate MGM was estimated at T1 and T4. Visualizations of estimated MGMs included varying edge thickness (thicker edges indicating larger edge weights), and different line types (solid = positive edge weights, dashed = negative edge weights). All models included the variables age at injury (years), sex (male/female), lesion level (C1 to S4-5 or intact) and severity (AIS grades A to E).

In terms of centrality indices, we investigated the expected influence and the bridge expected influence of each node in the estimated MGMs. The former estimates the connectivity of a node with all its neighboring nodes (by summing the shared edge weights) [36], the latter estimates the connectivity of a node with its neighboring nodes from other ICF domains than its own (by summing the shared edge weights between a node and its immediately connected nodes from other ICF domains) [37]. In addition to centrality, we examined the prevalence of functioning problems for each of the variables in the MGMs.

Network analysis results were investigated in terms of edge-weight accuracy and centrality stability by using different bootstrap approaches [34]. For edge-weight accuracy, non-parametric bootstrapping (N = 1000 samples of the data, with replacement) was used to estimate the 95% confidence intervals (CIs) for the edge weights. For the centrality stability, case-dropping bootstrapping (N = 1000 samples of the data) was used to estimate the correlation between the centrality indices of the original sample and those obtained from the bootstrapped subsamples. To assess the stability of centrality indices, their correlation stability coefficient can be calculated (CS-coefficient; preferably >0.5) [34].

Analyses were performed using RUMM2030 [38] and R version 4.2.2 for Windows [39]. Specifically, trajectory analysis was conducted using the R package lcmm version 2.0.0 [27], missing data imputation was performed using the R package missForest version 1.5 [29], network analysis was conducted using the R packages bootnet version 1.5 [34] and qgraph version 1.9.3 [40] for network estimation and visualization, respectively. A summary of the syntax for the final models of the trajectory and network analyses can be found in S1 Appendix.

Study reporting followed the checklist on Guidelines for Reporting on Latent Trajectory Studies [28] and the statement on Strengthening the Reporting of Observational Studies in Epidemiology [41].

## Results

In total, 1667 eligible individuals completed initial rehabilitation between May 2013 and March 2022, of whom 1488 consented to either the usage of routine clinical data for research purposes (N = 677) or the Inception Cohort (N = 811). Persons with implausible assessment time points of SCIM III (N = 4), with one or more SCIM III assessments before admission to initial rehab and during intensive care (N = 94), and none or only one assessment of the SCIM III during initial rehabilitation (N = 291) were excluded from this study, resulting in a total sample of 1099 participants. Descriptive information is shown in Table 1. The median age of participants was 58 years (IQR = 42–70 years), and the median time between SCI diagnosis and admission to initial rehabilitation was 14 days (IQR = 9–23 days). Most study participants were male (N = 744, 67.70%), with traumatic injury (N = 623, 56.69%), paraplegia (T1: N = 601, 54.69%), and incomplete injury (T1: N = 741, 67.42%).

### Rasch analysis

Good model fit (overall item fit residual SD = 0.76; overall person fit residual SD = 0.78; non-significant individual item fit residuals; non-significant chi-squared test statistic of item-trait interaction; PSI = 0.92) was achieved for the SCIM III total sum score by introducing two testlets: The first testlet included the items of the self-care and respiration and sphincter management subscales as well as the item about "*Mobility in bed*" from the mobility subscale. The second testlet included all remaining items of the mobility subscale. For this model, no DIF was present for etiology and sex, and only marginal DIF (<0.5) was found for age and time point of assessment. A similar good model fit and fulfilment of the assumptions of the Rasch measurement model was confirmed on the validation sample (see S4 Table).

**Table 1. Overview of available individuals who have consented to either the SwiSCI Inception Cohort or the use of routing clinical data for research purposes (MDS), as well as the excluded and included individuals for this study.**

| Characteristics | SwiSCI Inception Cohort or MDS (N = 1488) | Excluded from present study (N = 389) | Included within present study (N = 1099) | P-value |
|---|---|---|---|---|
| Sex = Female, n (%) | 481 (32.33) | 126 (32.39) | 355 (32.30) | 1.000 |
| Age at SCI diagnosis in years, median [IQR] | 58.00 [43.00, 71.00] | 60.00 [45.00, 72.00] | 58.00 [42.00, 70.00] | 0.081 |
| Etiology = Traumatic, n (%) | 847 (56.92) | 224 (57.58) | 623 (56.69) | 0.805 |
| Level of injury at T1, n (%) | | | | <0.001 |
| Tetraplegia | 489 (32.86) | 129 (33.16) | 360 (32.76) | |
| Paraplegia | 736 (49.46) | 135 (34.70) | 601 (54.69) | |
| Intact | 9 (0.60) | 2 (0.51) | 7 (0.64) | |
| Missing | 254 (17.07) | 123 (31.62) | 131 (11.92) | |
| Level of injury at T4, n (%) | | | | <0.001 |
| Tetraplegia | 458 (30.78) | 127 (32.65) | 331 (30.12) | |
| Paraplegia | 759 (51.01) | 164 (42.16) | 595 (54.14) | |
| Intact | 30 (2.02) | 12 (3.08) | 18 (1.64) | |
| Missing | 241 (16.20) | 86 (22.11) | 155 (14.10) | |
| Severity of injury at T1, n (%) | | | | <0.001 |
| AIS A | 285 (19.15) | 72 (18.51) | 213 (19.38) | |
| AIS B | 142 (9.54) | 28 (7.20) | 114 (10.37) | |
| AIS C | 174 (11.69) | 47 (12.08) | 127 (11.56) | |
| AIS D | 619 (41.60) | 119 (30.59) | 500 (45.50) | |
| AIS E | 8 (0.54) | 2 (0.51) | 6 (0.55) | |
| Missing | 260 (17.47) | 121 (31.11) | 139 (12.65) | |
| Severity of injury at T4, n (%) | | | | <0.001 |
| AIS A | 245 (16.47) | 71 (18.25) | 174 (15.83) | |
| AIS B | 93 (6.25) | 13 (3.34) | 80 (7.28) | |
| AIS C | 108 (7.26) | 23 (5.91) | 85 (7.73) | |
| AIS D | 755 (50.74) | 180 (46.27) | 575 (52.32) | |
| AIS E | 29 (1.95) | 12 (3.08) | 17 (1.55) | |
| Missing | 258 (17.34) | 90 (23.14) | 168 (15.29) | |
| Length of stay in days, median [IQR] | 135.00 [74.00, 191.00] | 124.50 [49.00, 195.00] | 139.00 [80.00, 190.00] | 0.001 |
| Missing, n (%) | 3 (0.20) | 3 (0.77) | 0 (0.00) | |
| Interval-based SCIM III sum score at T1, median [IQR] | 74.60 [58.33, 87.42] | 40.25 [12.30, 68.41] | 75.93 [62.75, 87.77] | <0.001 |
| Missing, n (%) | 317 (21.30) | 264 (67.87) | 53 (4.82) | |
| Interval-based SCIM III sum score at T2, median [IQR] | 85.98 [74.60, 94.07] | 68.41 [50.38, 83.84] | 86.97 [77.20, 94.42] | <0.001 |
| Missing, n (%) | 911 (61.22) | 341 (87.66) | 570 (51.87) | |
| Interval-based SCIM III sum score at T3, median [IQR] | 84.92 [71.67, 91.72] | 80.45 [56.50, 84.77] | 86.46 [74.60, 92.70] | 0.002 |
| Missing, n (%) | 1196 (80.38) | 351 (90.23) | 845 (76.89) | |
| Interval-based SCIM III sum score at T4, median [IQR] | 94.42 [84.92, 97.30] | 90.98 [75.93, 97.27] | 94.42 [85.98, 97.30] | 0.002 |
| Missing, n (%) | 204 (13.71) | 192 (49.36) | 12 (1.09) | |
| Assessment time point SCIM III T1 in days, median [IQR] | 9.00 [1.00, 19.00] | -8.50 [-21.25, 1.00] | 11.00 [2.00, 19.00] | <0.001 |
| Missing, n (%) | 314 (21.10) | 261 (67.10) | 53 (4.82) | |
| Assessment time point SCIM III T2 in days, median [IQR] | 67.00 [55.00, 76.00] | 44.00 [17.75, 54.75] | 69.00 [57.00, 77.00] | <0.001 |
| Missing, n (%) | 911 (61.22) | 341 (87.66) | 570 (51.87) | |

(*Continued*)

**Table 1.** (Continued)

| Characteristics | SwiSCI Inception Cohort or MDS (N = 1488) | Excluded from present study (N = 389) | Included within present study (N = 1099) | P-value |
|---|---|---|---|---|
| Assessment time point SCIM III T3 in days, median [IQR] | 145.00 [131.00, 158.00] | 129.00 [111.25, 140.75] | 147.00 [134.00, 159.00] | <0.001 |
| Missing, n (%) | 1196 (80.38) | 351 (90.23) | 845 (76.89) | |
| Assessment time point SCIM III T4 in days, median [IQR] | 133.00 [71.25, 188.00] | 125.00 [34.50, 201.50] | 135.00 [76.00, 186.50] | 0.060 |
| Missing, n (%) | 202 (13.58) | 190 (48.84) | 12 (1.09) | |

Variable distributions between excluded and included samples were compared using p-values of the Mann-Whitney-U-Test for continuous variables and the Pearson's Chi-square test for categorical variables (both with continuity correction). Abbreviations: AIS, American Spinal Injury Association Impairment Scale; IQR, Interquartile range; MDS, minimal data set; SCIM III, Spinal Cord Independence Measure version III; SwiSCI, Swiss Spinal Cord Injury Cohort Study; T1-T4, SwiSCI assessment time points 1–4. A more detailed overview is given in S3 Table.

Pearson correlation coefficients confirmed the robustness of the SCIM III total scores compared to the previous analysis [24] and were r = 0.99 at T1 (N = 769) and T4 (N = 858), and r = 0.98 at T2 (N = 387) and T3 (N = 198).

No model fit was achieved for the corresponding analyses of the SCIM III subscales. Corresponding results are available from the authors on request.

## Trajectory analysis

Observed individuals' functioning trajectories, modelled with the SCIM III total score, during initial rehabilitation are shown in S1 Fig: Two SCIM III assessments were available for 598 participants, three assessments for 284 participants, and four assessments for 217 participants.

Considering that no good fit was achieved for the Rasch Analysis of the SCIM III subscales, we focused on the univariate trajectory analyses using the SCIM III total score as outcome. The corresponding analysis of suitable parameterized link functions can be found in S2 Fig. According to the AIC, the quadratic I-splines function with three knots was identified as best-fitting. However, similar to previous results [6], the quadratic I-splines function with two knots was selected as the best-fitting link function in order to minimize overfitting. The fit characteristics of the subsequent trajectory analyses are shown in Table 2 for both model specification sets (fixed vs. varying between-person trajectory variability across classes) with increasing class enumeration. For a visual interpretation of the fitted models, class-specific predicted mean functioning trajectories are shown in S3 and S4 Figs for the two model sets respectively. Both model sets showed similar results and a good visual interpretability in terms of predicted mean functioning trajectories. Due to its greater flexibility, and similar to previous results [6], we preferred the model set 2, which allows the variability of trajectories between individuals to vary across classes, over set 1.

In the model set 2, the fit indices indicated two different models: The 4-class model was preferred by BIC, while AIC and SABIC preferred the 5-class model. With a preference for BIC over SABIC and AIC, the 4-class model was considered as best-fitting, even though it included one class that represented less than 5% of the total sample size. In addition, this solution had a good entropy value of 0.80. The correspondingly identified predicted mean functioning trajectories for each class are summarized in Fig 1 and describe similar patterns to the original analysis [6]: *Stable high functioning* (N = 239; 21.75%), *early* (N = 33; 3.00%), *moderate* (N = 753; 68.52%), and *slow* (N = 74; 6.73%) *functioning improvement*, respectively. Posterior classification accuracy is presented in S5 Table and shows that the mean misclassification rates was

**Table 2. Fit characteristics of the latent process mixed models.**

| Model specifications | No. of classes | Random starts (departures/ iterations) | No. of model parameters | BIC | SSABIC | AIC | Entropy | Class 1 sample size, n (%) | Class 2 sample size, n (%) | Class 3 sample size, n (%) | Class 4 sample size, n (%) | Class 5 sample size, n (%) | Class 6 sample size, n (%) |
|---|---|---|---|---|---|---|---|---|---|---|---|---|---|
| Set 1: LPMMs with *fixed* between-person trajectory variability across classes | 1 | 200/100 | 8 | 22131.55 | 22106.14 | 22091.53 | 1.00 | 1099 (100.00) | | | | | |
| | 2 | 200/100 | 11 | 22089.87 | 22054.94 | 22034.85 | 0.74 | 922 (83.89) | 177 (16.11) | | | | |
| | 3 | 200/100 | 14 | 22050.81 | 22006.35 | 21980.78 | 0.71 | 511 (46.50) | 48 (4.37) | 540 (49.14) | | | |
| | 4 | 200/100 | 17 | 21971.35 | 21917.35 | 21886.31 | 0.76 | 68 (6.19) | 438 (39.85) | 449 (40.86) | 144 (13.10) | | |
| | 5 | 200/100 | 20 | 21962.30 | 21898.78 | 21862.26 | 0.76 | 86 (7.83) | 72 (6.55) | 365 (33.21) | 404 (36.76) | 172 (15.65) | |
| | 6 | 200/100 | 23 | 21962.37 | 21889.32 | 21847.32 | 0.77 | 57 (5.19) | 266 (24.20) | 328 (29.85) | 86 (7.83) | 226 (20.56) | 136 (12.37) |
| Set 2: LPMMs with *varying* between-person trajectory variability across classes | 1 | 250/200 | 8 | 22131.55 | 22106.14 | 22091.53 | 1.00 | 1099 (100.00) | | | | | |
| | 2 | 250/200 | 12 | 22077.45 | 22039.33 | 22017.42 | 0.88 | 992 (90.26) | 107 (9.74) | | | | |
| | 3* | 250/200 | 16 | 21930.22 | 21879.40 | 21850.19 | 0.80 | 30 (2.73) | 829 (75.43) | 240 (21.84) | | | |
| | **4** | **250/200** | **20** | **21898.26** | **21834.74** | **21798.22** | **0.80** | **33 (3.00)** | **239 (21.75)** | **74 (6.73)** | **753 (68.52)** | | |
| | 5 | 250/200 | 24 | 21905.73 | 21829.50 | 21785.68 | 0.72 | 55 (5.00) | 525 (47.77) | 239 (21.75) | 70 (6.37) | 210 (19.11) | |
| | 6* | 250/200 | 28 | 21908.57 | 21819.63 | 21768.51 | 0.74 | 59 (5.37) | 212 (19.29) | 70 (6.37) | 233 (21.20) | 524 (47.68) | 1 (0.09) |

The best-fitting latent process mixed model is marked in bold. Abbreviations: AIC, Akaike information criterion; BIC, Bayesian information criterion; LPMM, latent process mixed model; SSABIC, sample-size adjusted Bayesian information criterion.

*Convergence problems.

highest for the *early improvement class* with a 21.05% probability of being classified as *moderate improvement*. Sample characteristics for each class are summarized in Table 3, and the detailed estimated parameters of the model are given in S6 Table.

## Network analysis

Due to the small sample size of some of the classes identified in the trajectory analysis, network analysis was performed only for the *stable high* (N = 239) and the *moderate improvement classes* (N = 753). Of these, the accuracy analysis of the edge weights showed higher accuracy for the *moderate improvement class* and lower accuracy for the *stable high class*. Therefore, the following results focus on the *moderate improvement class*, and the corresponding prevalence of functioning problems is shown in Fig 2. The corresponding results for the *stable high class* are available from the authors on request.

For all MGMs, the EBIC hyperparameter was manually set to 0 to allow for exploratory model estimation (all estimated EBIC hyperparameters showed similar models and corresponding results are available from the authors upon request). The estimated MGMs at T1 and T4 are shown in Fig 3 and the absolute edge weights of the networks range from 0.04 to 0.64 at T1 and from 0.05 to 0.56 at T4 (see S8 Table). Three relatively strong positive edges are present

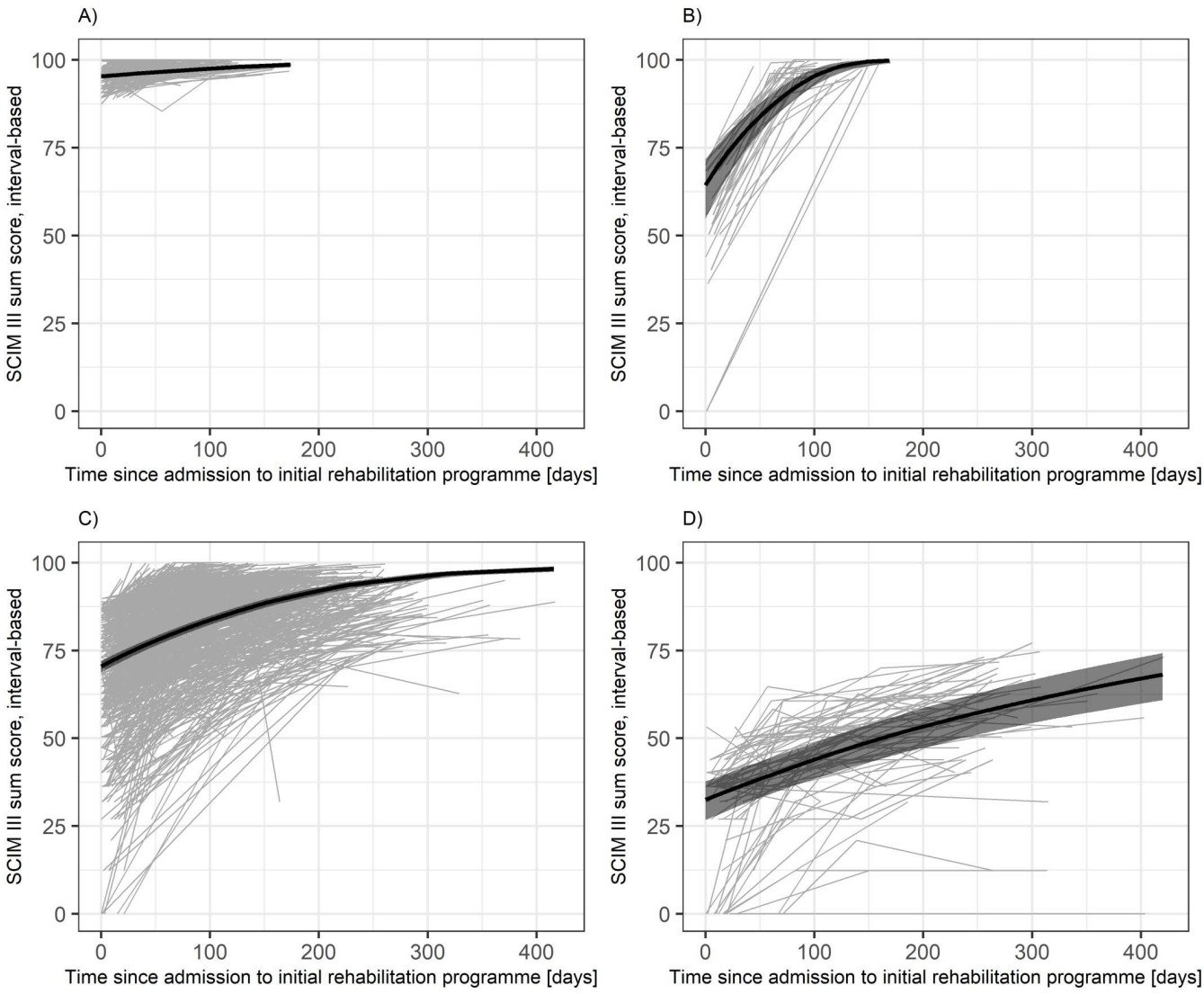

**Fig 1. Predicted mean (black) with 95% confidence intervals and observed (grey) functioning trajectories according to the best-fitting latent process mixed model.** A) stable high functioning class (N = 239; 21.75%). B) early functioning improvement class (N = 33; 3.00%). C) moderate functioning improvement class (N = 753; 68.52%). D) slow functioning improvement class (N = 74; 6.73%). Trajectories were plotted up to the maximum assessment time point of each class observed in the study sample.

for the *moderate improvement class* at both time points: "*Mobility indoors–Mobility moderate distances*", "*Bathing lower body–Dressing lower body*", and "*Grooming–Feeding*". On the other hand, the node "*Respiration*" is hardly connected at any time point. Generally, the nodes are roughly clustered around the SCIM subscales: variables of self-care ("*Eating/ Drinking*", "*Washing oneself/ Caring for body parts*", "*Dressing*"), sphincter management ("*Toileting*") and mobility (*"Changing basic body position/ Transferring oneself", " Walking/ Moving around/ Moving around using equipment"*) are mainly connected with other variables of the same sub-scale. In addition, exceptional nodes such as "*Mobility in bed*" and "*Use of toilet*" act as bridges between variables from different SCIM subscales, as they mainly support connections to vari-ables from other subscales at both time points. The estimated edge weight accuracy and corre-sponding bootstrapped CIs are shown in S5 Fig. The figure shows the sample and bootstrap

**Table 3. Overview of study participant characteristics for the identified classes of functioning trajectories.**

| Characteristics | Stable high functioning class | Early functioning improvement class | Moderate functioning improvement class | Slow functioning improvement class |
|---|---|---|---|---|
| | (N = 239) | (N = 33) | (N = 753) | (N = 74) |
| Sex = Female, n (%) | 78 (32.64) | 12 (36.36) | 251 (33.33) | 14 (18.92) |
| Age at SCI diagnosis in years, median [IQR] | 51.00 [37.00, 60.50] | 61.00 [46.00, 69.00] | 60.00 [44.00, 72.00] | 65.50 [43.25, 74.00] |
| Etiology = Traumatic, n (%) | 132 (55.23) | 22 (66.67) | 426 (56.57) | 43 (58.11) |
| Level of injury at T1, n (%) | | | | |
| Tetraplegia | 71 (29.71) | 16 (48.48) | 211 (28.02) | 62 (83.78) |
| Paraplegia | 121 (50.63) | 9 (27.27) | 466 (61.89) | 5 (6.76) |
| Intact | 5 (2.09) | 0 (0.00) | 2 (0.27) | 0 (0.00) |
| Missing | 42 (17.57) | 8 (24.24) | 74 (9.83) | 7 (9.46) |
| Level of injury at T4, n (%) | | | | |
| Tetraplegia | 63 (26.36) | 15 (45.45) | 195 (25.90) | 58 (78.38) |
| Paraplegia | 120 (50.21) | 13 (39.39) | 458 (60.82) | 4 (5.41) |
| Intact | 10 (4.18) | 1 (3.03) | 7 (0.93) | 0 (0.00) |
| Missing | 46 (19.25) | 4 (12.12) | 93 (12.35) | 12 (16.22) |
| Severity of injury at T1, n (%) | | | | |
| AIS A | 5 (2.09) | 1 (3.03) | 178 (23.64) | 29 (39.19) |
| AIS B | 9 (3.77) | 0 (0.00) | 93 (12.35) | 12 (16.22) |
| AIS C | 4 (1.67) | 5 (15.15) | 98 (13.01) | 20 (27.03) |
| AIS D | 172 (71.97) | 20 (60.61) | 302 (40.11) | 6 (8.11) |
| AIS E | 4 (1.67) | 0 (0.00) | 2 (0.27) | 0 (0.00) |
| Missing | 45 (18.83) | 7 (21.21) | 80 (10.62) | 7 (9.46) |
| Severity of injury at T4, n (%) | | | | |
| AIS A | 4 (1.67) | 1 (3.03) | 147 (19.52) | 22 (29.73) |
| AIS B | 9 (3.77) | 0 (0.00) | 62 (8.23) | 9 (12.16) |
| AIS C | 0 (0.00) | 0 (0.00) | 66 (8.76) | 19 (25.68) |
| AIS D | 166 (69.46) | 27 (81.82) | 371 (49.27) | 11 (14.86) |
| AIS E | 9 (3.77) | 1 (3.03) | 7 (0.93) | 0 (0.00) |
| Missing | 51 (21.34) | 4 (12.12) | 100 (13.28) | 13 (17.57) |
| Length of stay in days, median [IQR] | 59.00 [42.00, 85.50] | 105.00 [92.00, 128.00] | 165.00 [113.00, 201.00] | 246.50 [190.50, 276.50] |
| Interval-based SCIM III sum score at T1, median [IQR] | 95.96 [94.07, 97.30] | 62.75 [50.38, 68.41] | 73.18 [62.75, 81.39] | 36.29 [27.02, 40.25] |
| Missing, n (%) | 0 (0.00) | 7 (9.46) | 32 (4.25) | 14 (5.86) |
| Interval-based SCIM III sum score at T2, median [IQR] | 97.27 [95.96, 98.31] | 94.95 [91.00, 97.30] | 85.98 [78.36, 91.53] | 43.91 [36.29, 53.23] |
| Missing, n (%) | 14 (42.42) | 31 (41.89) | 343 (45.55) | 182 (76.15) |
| Interval-based SCIM III sum score at T3, median [IQR] | 96.97 [96.14, 98.54] | 95.87 [94.97, 96.77] | 88.33 [78.36, 94.07] | 50.38 [43.91, 60.61] |
| Missing, n (%) | 31 (93.94) | 33 (44.59) | 547 (72.64) | 234 (97.91) |
| Interval-based SCIM III sum score at T4, median [IQR] | 98.54 [97.27, 99.62] | 97.58 [96.01, 98.93] | 91.72 [85.45, 95.96] | 54.56 [43.91, 62.75] |
| Missing, n (%) | 1 (3.03) | 0 (0.00) | 7 (0.93) | 4 (1.67) |
| Assessment time point SCIM III T1 in days, median [IQR] | 10.00 [2.00, 20.00] | 7.00 [2.00, 17.00] | 11.00 [2.00, 19.00] | 14.00 [5.00, 23.00] |
| Missing, n (%) | 0 (0.00) | 7 (9.46) | 32 (4.25) | 14 (5.86) |
| Assessment time point SCIM III T2 in days, median [IQR] | 56.00 [43.00, 71.00] | 69.00 [61.50, 78.50] | 69.00 [59.00, 77.00] | 71.00 [62.50, 82.00] |
| Missing, n (%) | 14 (42.42) | 31 (41.89) | 343 (45.55) | 182 (76.15) |

*(Continued)*

**Table 3.** (Continued)

| Characteristics | Stable high functioning class | Early functioning improvement class | Moderate functioning improvement class | Slow functioning improvement class |
|---|---|---|---|---|
| | (N = 239) | (N = 33) | (N = 753) | (N = 74) |
| Assessment time point SCIM III T3 in days, median [IQR] | 91.00 [27.00, 148.00] | 136.50 [134.25, 138.75] | 147.00 [134.25, 159.00] | 150.00 [139.00, 161.00] |
| Missing, n (%) | 31 (93.94) | 33 (44.59) | 547 (72.64) | 234 (97.91) |
| Assessment time point SCIM III T4 in days, median [IQR] | 55.00 [38.50, 82.50] | 102.00 [91.00, 126.75] | 161.00 [108.00, 195.00] | 243.00 [187.25, 268.75] |
| Missing, n (%) | 1 (3.03) | 0 (0.00) | 7 (0.93) | 4 (1.67) |

Abbreviations: AIS, American Spinal Injury Association Impairment Scale; SCIM III, Spinal Cord Independence Measure version III; SwiSCI, Swiss Spinal Cord Injury Cohort Study; T1-T4, SwiSCI assessment time points 1–4. A more detailed overview is given in S7 Table.

mean edge weights and 95% CIs. The overlapping bootstrapped CIs indicate that the order of most of the edge weights in the networks should be interpreted with caution.

The estimated centrality indices of expected influence and bridge expected for both networks of the *moderate improvement class* are shown in Figs 4 and 5, respectively (see S9 and S10 Tables for detailed estimates). The figures show the sample and bootstrap mean centrality indices and bootstrapped 95% CIs. Again, overlapping bootstrapped CIs indicate that the order of the indices should be interpreted with caution. In terms of the expected influence index, the most influential nodes mainly cover variables of the self-care and mobility subscales, and less influential nodes are "*Respiration*" and individual variables of the mobility and sphincter management subscales. In terms of the bridge expected influence index, the most influential nodes are dominated by variables of the self-care subscale, and less influential nodes are again "*Respiration*" and individual mobility and sphincter management variables. The bootstrapped stability estimates of the centrality indices can be found in S6 Fig. They show that the overall stability of both indices is good and comparable at T1 and T4, with some increasing uncertainty for the bridge expected influence at T4. This is supported by the estimated CS-coefficients for the expected influence (0.75 at T1 and *T4)* and the bridge expected influence (0.75 at T1 and T4).

Combining the information from the two centrality measures at T1, we can see that for the self-care subscale, "*Feeding*", "*Dressing upper body*", and "*Dressing lower body*" are all among the most influential nodes for both indices. For the mobility and the sphincter management subscales, only "*Mobility in bed*" and "*Use of toilet*" are among the most influential nodes for both indices, respectively. In addition, the nodes "*Dressing lower body*", "*Use of toilet*" and "*Mobility in bed*" are highly prevalent functioning problems in the *moderate improvement class* with 77.95%, 69.49%, and 50.07% of the sample reporting the need for maximal assistance at T1, respectively. And they are directly related to other highly prevalent functioning problems, such as "*Bathing lower body*" and "*Bowel management*" with 71.98% and 45.77% of the sample reporting the need for maximal assistance at T1, respectively.

Finally, a comparison of the networks between T1 and T4 in Fig 3 shows that for variables from the sphincter management and mobility subscales, the number of connections between corresponding nodes increases, and the variables thus become more connected. This is also reflected in the centrality measures: While in terms of the bridge expected influence, variables from the selfcare subscale tend to remain the most influential at T4, along with "*Use of toilet*" and "*Mobility in bed*"; in terms of the expected influence, variables from the sphincter management and mobility subscales tend to become more influential on overall functioning at T4.

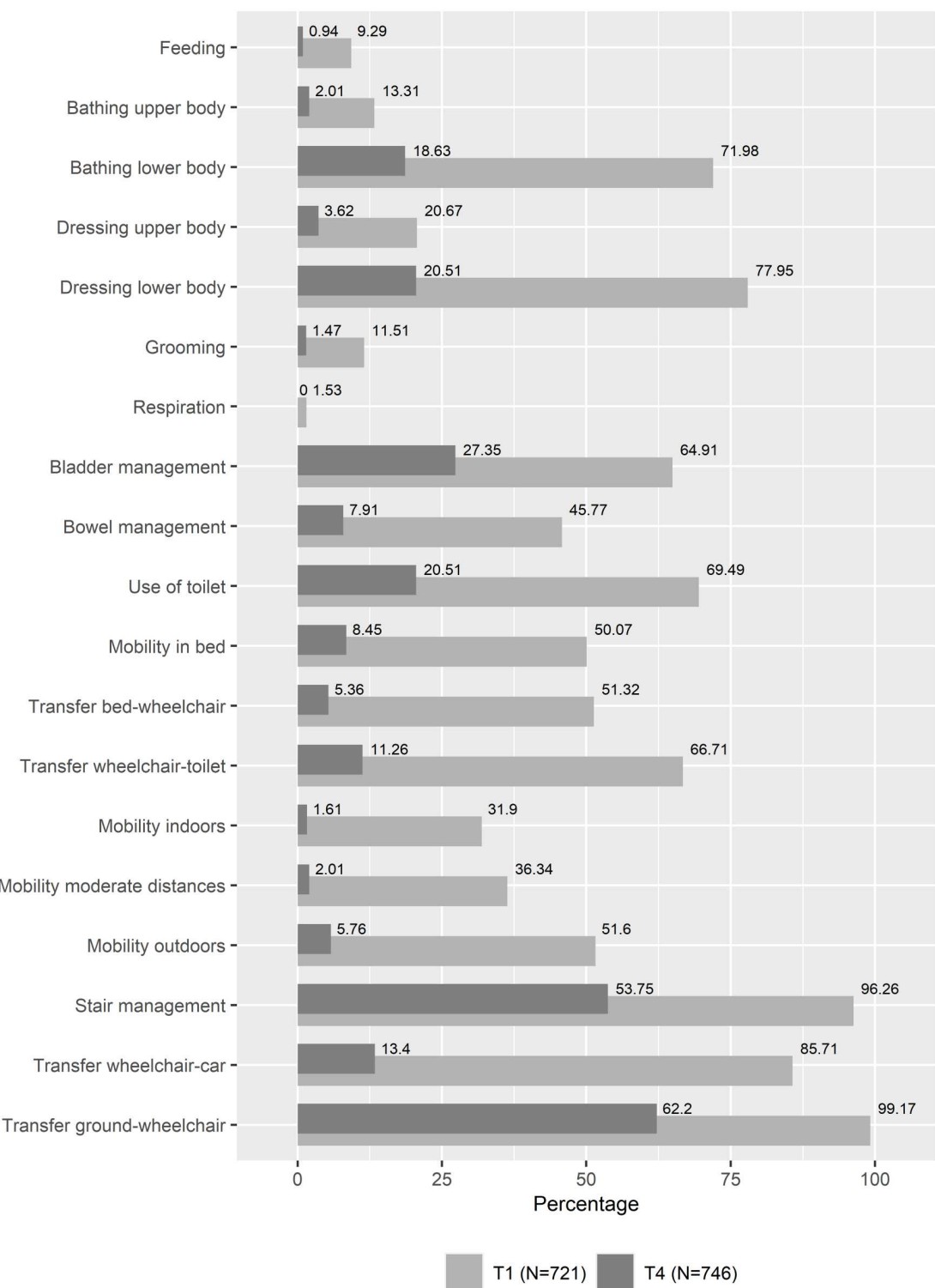

**Fig 2. Prevalence of functioning problems for the moderate functioning improvement class according to SCIM III variables.** For each SCIM III variable, the percentage of participants who reported the need for maximal assistance is shown (i.e., response category 0).

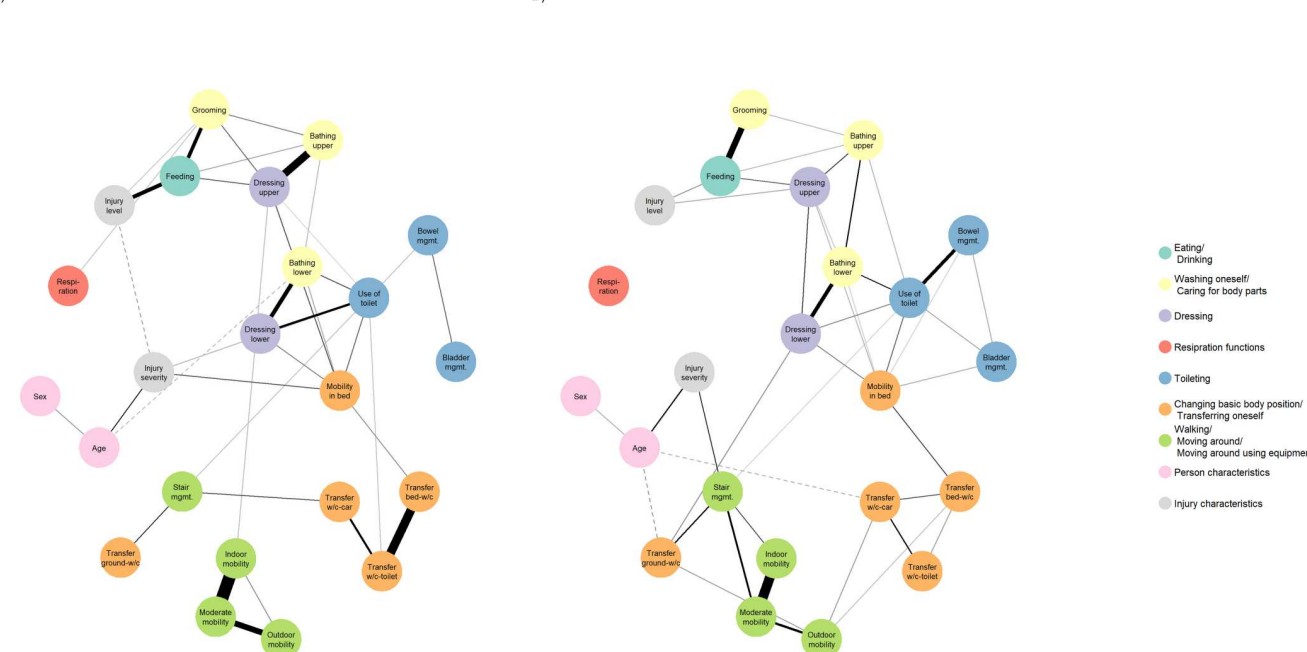

**Fig 3. Estimated mixed graphical model networks for the moderate functioning improvement class.** A) T1 (N = 721). B) T4 (N = 746). Sex is interpreted as a binary variable with male as the reference group. All other variables are interpreted as continuous (ordered) variables: For the SCIM III items, higher scores indicate higher independence in performing ADLs; for age at injury, higher scores indicate older age; and for injury level and severity, higher scores indicate lower lesion level and severity, respectively. The networks were plotted using an averaged layout based on the Fruchterman-Reingold algorithm, which places two nodes closer together for stronger edge weights between them.

## Discussion

This study re-estimated previously published functioning trajectory models [6] based on an updated dataset, and confirmed the corresponding four trajectory classes of *stable high functioning* (N = 239; 21.75%), *early* (N = 33; 3.00%), *moderate* (N = 753; 68.52%), and *slow* (N = 74; 6.73%) *functioning improvement*, respectively. Corresponding association structures between relevant functioning problems in the *moderate improvement class* were explored and centrality indices at T1 revealed "*Feeding*", "*Dressing upper body*", and "*Dressing lower body*" in the self-care subscale, "*Mobility in bed*" in the mobility subscale, and "*Use of toilet*" in the respiration and sphincter management subscale as highly connected nodes in the network for both indices. These functioning domains might indicate potential trajectory-specific targets for clinical interventions.

### Trajectory analysis

The mean functioning trajectories of persons with SCI during rehabilitation are similar to available evidence but some differences were also observed. The trajectory analysis revealed similar but changing shapes and class membership proportions of the identified mean functioning trajectories compared to the previous analysis. However, contrary to our previous hypothesis [6] that the previous *slow improvement class* might split into two separate classes with increasing sample size, we observed that the *moderate improvement class* absorbed parts of the previous *slow improvement* and *stable high classes*. Thus, in the re-estimated analysis, the *moderate improvement class* became more heterogeneous than before. The non-splitting of the

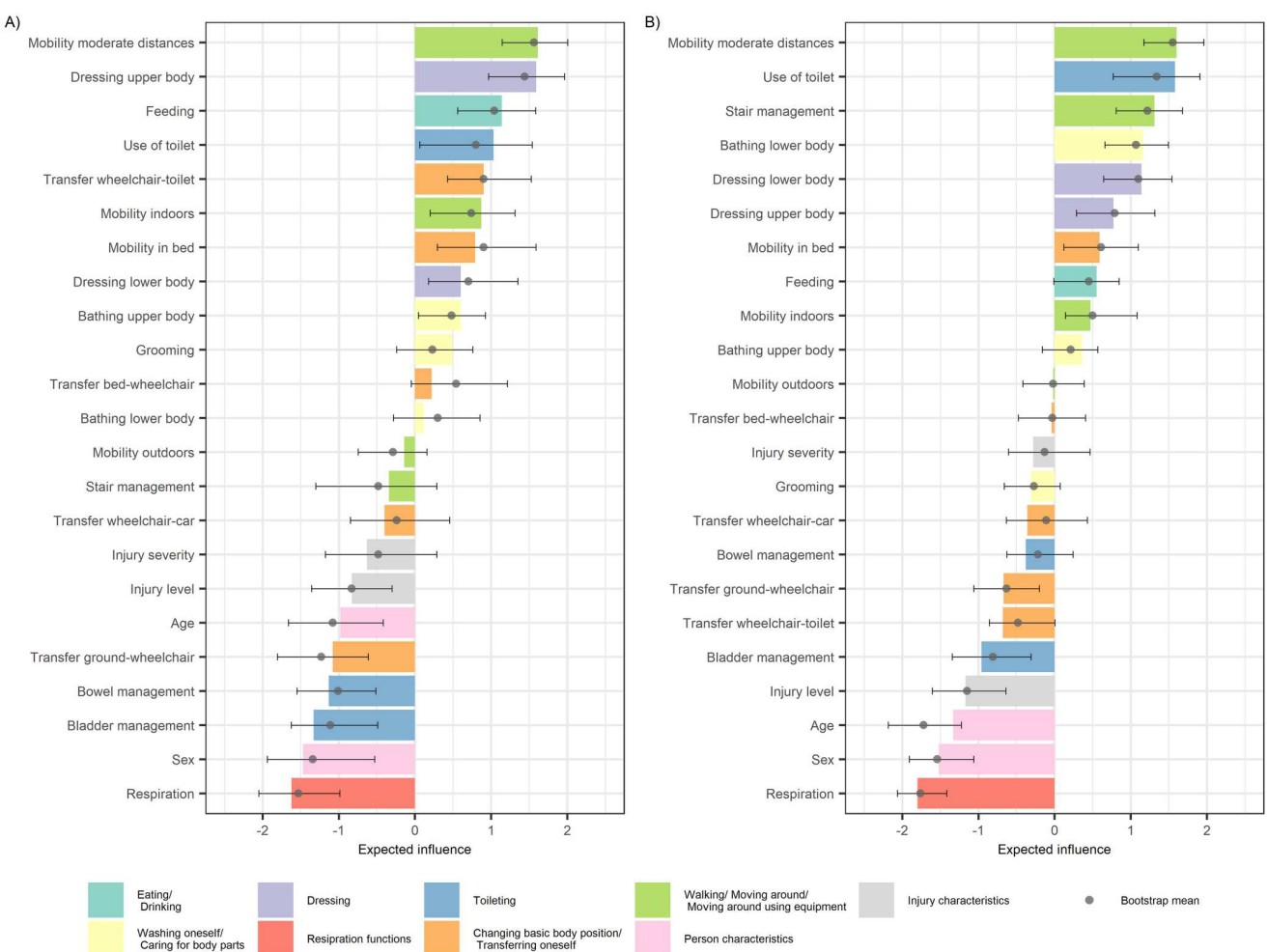

**Fig 4. Estimated expected influence (z-scores) for each variable based on the mixed graphical model networks for the moderate improvement class.**
A) T1. B) T4. The color bars indicate the estimated indices based on the original sample of each network; the grey dots and error bars indicate the bootstrap means and corresponding 95% bootstrap confidence intervals, respectively.

*slow improvement class* could be explained by the fact that the updated dataset still contains few (very) low individual functioning trajectories. Although the re-estimation of the results from the previous analysis confirmed the number of the trajectories, the decision between the 4-class and the 5-class solution was ambiguous (see Table 2 and S4 Fig): Since the 4-class solution includes a small class (N<5% of the sample size) and the 5-class solution shows interpretable mean functioning trajectories, an argumentation in favor of the 5-class solution as best-fitting would have been possible. However, Weller et al. [42] have reported that the preference for class sample sizes >5% of the study sample has been relaxed in practice, if the small class is supported conceptually and by other model fit statistics. In our view, the ambiguous decision between the 4-class and 5-class solutions, the more heterogeneous *moderate improvement class*, as well as the few low individual functioning trajectories in the dataset, might point to the emergence of a 5th or 6th class in analyses with even larger sample sizes and more balanced datasets. In addition, as mentioned by Tulsky et al [7], the large within-class variability that is overserved for some classes makes classification and interpretation difficult at the individual level. Consequently, they turned to an individual growth curve analysis approach to investigate

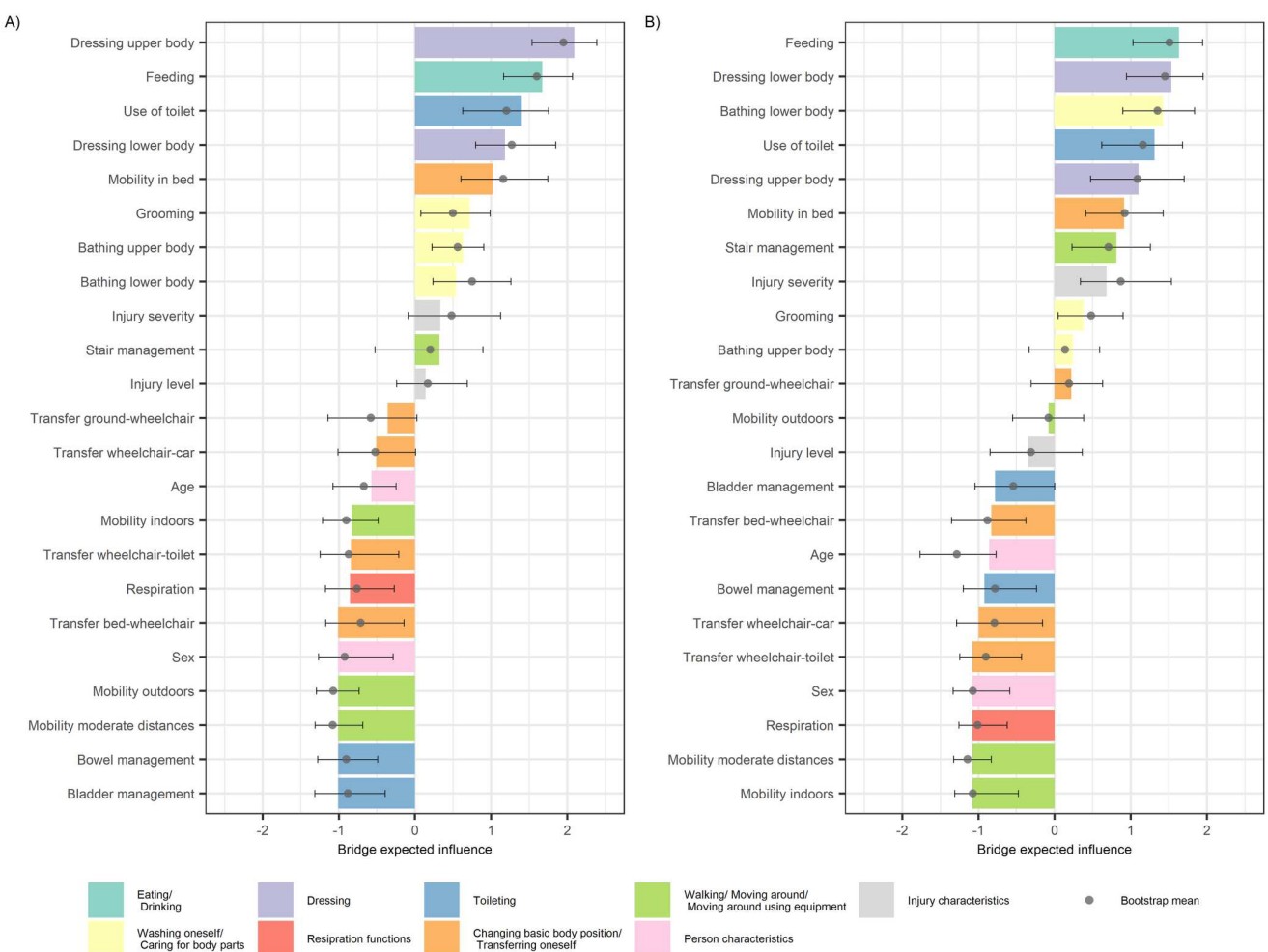

**Fig 5. Estimated bridge expected influence (z-scores) for each variable based on the mixed graphical model networks for the moderate functioning improvement class.** A) T1. B) T4. The color bars indicate the estimated indices based on the original sample of each network; the grey dots and error bars indicate the bootstrap means and corresponding 95% bootstrap confidence intervals, respectively.

trajectory predictors [7]. Thus, for the potential use of functioning trajectories in clinical practice (e.g., the development of corresponding clinical prediction models), it will be particularly important to critically evaluate and discuss the heterogeneity of the identified classes in a given context (e.g., who could make use of the trajectories, for what purpose and in which situations).

From a clinical perspective, a closer examination of the differences between the *early* and the *moderate improvement classes* may be worthwhile, as the mean trajectories of the two classes start at about the same functioning scores, but show quite different trajectories thereafter. There are several possible explanations for this: The two classes differ in their composition in terms of both the level and severity of injury, with the *early improvement class* including a majority of persons with tetraplegia and AIS grade D, whereas the *moderate improvement class* includes a relatively higher proportion of persons with paraplegia and AIS grades A and B at T1 (Table 3). Other compositional differences include higher proportions for pre-SCI comorbidities and the presence of a pressure injury at T1 for the *moderate improvement class*, and higher proportions for the presence of a partner at the time of SCI and normal defecation at

T1 for the *early improvement class* (S7 Table). However, these proportions should be interpreted with caution due to the high number of missing observations for some variables. Nevertheless, this opens the door for further investigation into potential modifiable factors associated with the *moderate improvement class* that could improve functioning outcomes early in rehabilitation to support faster recovery.

The operationalization of functioning for the estimation of trajectories has an important impact on results. In our study, the SCIM III was used to operationalize functioning because it is widely used in clinical practice. However, although previous Rasch analyses of different versions of the SCIM exist [43–45], it was originally developed without Rasch analysis [46]. In addition, due to the qualitative development of functioning after complete or incomplete SCI concerning bladder function for example, the response categories of the SCIM III are not strictly ordinal for some items, but rather categorical. And the non-consecutive scoring of some items due to assumed clinical relevance seems arbitrary. Therefore, the SCIM III may not be best suited to construct a metrical scale of functioning with the Rasch measurement model and, in fact, the introduction of item testlets was necessary to meet all the assumptions of the Rasch model. In addition, the SCIM III lacks important domains of functioning included in the Brief ICF Core Set for SCI, such as cognitive functions or communication, potentially resulting in an uncomplete picture of an individual's overall functioning status. Finally, it is prone to floor and ceiling effects [44, 47], making the interpretation of very high or low functioning trajectories difficult. As some concerns regarding the SCIM III have recently been addressed in the development of the SCIM version IV [45] (e.g., internal consistency, floor effects, wording and scoring of some items), future research needs to examine the consequences for the presented Rasch and trajectory analysis based on this updated version of the SCIM. And ideally, functioning measures and scores other than the SCIM should be considered as alternatives to independently confirm and contrast the trajectory classes identified in this study.

## Network analysis

To our knowledge, this is one of the first studies implementing network analysis to examine complex association structures among functioning problems of persons with SCI undergoing initial rehabilitation. Similar approaches have been described by Strobl et al [30] in early post-acute rehabilitation but not specific to the SCI population. Kalisch et al [48], Reinhardt et al [49], and Ehrmann et al [50–52] have examined functioning association structures in persons with SCI in either the community or mixed settings (early post-acute and long-term situation) in Switzerland or other countries. However, the comparability of our findings with the results of other studies is limited because different studies often use a different variable set for network analysis. In addition, other studies either used no centrality indices or centrality indices other than the expected and bridge expected influence indices.

The combination of information from centrality measures and prevalence unveils impactful functioning problems and can inform potential targets for interventions. The estimated measure of centrality and prevalence do not always indicate the same variables as being of particular interest. Some variables appear to be highly prevalent functioning problems, for example "*Transfer ground-wheelchair*" at T1 in the *moderate improvement class*, but are not very central. This is also true for the two centrality measures as such. In addition, the bridge expected influence is, by definition, based on the specified grouping of functioning variables into ICF domains, whereas the expected influence is not. Therefore, the bridge expected influence is sensitive to changes in the definition or selection of the ICF domains represented by the functioning variables. Thus, we have attempted to combine various centrality indices together with

problem prevalence to provide a more comprehensive and robust picture of potential interesting targets for intervention rather than simply relying on a single measure. Specifically, "*Feeding*", "*Dressing upper body*", and "*Dressing lower body*" (self-care subscale), "*Mobility in bed*" (mobility subscale) and "*Use of toilet*" (sphincter management subscale) have been identified among the most influential nodes for both indices, respectively. In addition, we observed that among these nodes, "*Feeding*" and "*Dressing upper body*", as well as "*Dressing lower body*", "*Use of Toilet*" and "*Mobility in bed*" are directly connected (Fig 3). Therefore, interventions that can simultaneously target such connected patterns of influential nodes may be of particular interest to improve overall functioning at T1. However, from Figs 2 and 3 it is clear that the variables of the mobility subscale, which mostly also show a high prevalence of functioning problems at T1, are only weakly related to the remaining other nodes. Therefore, specific interventions targeting problems in the variables "*Mobility moderate distances*" and "*Transfer wheelchair-toilet*" may be of complementary interest to specifically influence overall functioning through general mobility. Finally, the comparison of networks between T1 and T4 (Fig 3) revealed changing connectivity between specific variables and ICF domains. Although these shifts in terms of centrality indices may reflect changing therapy and rehabilitation goals during initial rehabilitation between T1 and T4, a direct comparison of the estimated networks with, for example, existing rehabilitation process maps to contextualize our results proved to be difficult because of the different levels of detail and timing, and because such maps were not developed specifically for trajectory classes, but rather for injury levels and/or severity grades.

Centrality indices should be understood as a tool to summarize the association of a variable in a network according to the strength of its connections (edges). In this sense, they can point to interesting areas of functioning as targets for interventions, but they do not reflect interventions and their effects as such. Since edges in a network do not necessarily reflect causal relationships but may also indicate bidirectional or conditional relationships between nodes, or the presence of unmodelled latent variables [33], centrality indices in turn do not necessarily assess the impact caused by specific rehabilitation interventions in clinical practice. Therefore, our results must be interpreted with caution: Centrality measures give no indication on clinical importance of one variable over the other, best timing, or prioritization of intervention targets. Most importantly and as highlighted by Santos et al for psychological networks [53], it is possible that non-central nodes are highly clinically relevant and are essentially related to a person's overall functioning and well-being. Generally, the interpretation and use of centrality indices, for example in psychological networks [54], is debated and depends on the specific purpose of a study. Therefore, the presented results need to be used in combination with clinical information and expertise. Despite these caveats, network analysis can be an important approach to complement the information from ICF Core Sets: Using the ICF Core Set for SCI as a reference framework, we were able to ensure that selected functioning variables in the network analysis represent relevant functioning problems in persons with SCI in the early post-acute situation. Having this as a starting point, network analysis allowed us to identify potential impactful functioning categories to be considered in clinical rehabilitation.

Future studies are needed to confirm our results, to extend the analysis to the *early* and *slow functioning improvement classes* and to investigate relevant functioning problems that could not be taken into account in our network analysis approach. Although the use of the ICF Brief Core Set for SCI ensured the relevance of selected functioning variables, we were unable to include some of the functioning categories represented in the Core Set in our network analysis due to the additional selection criteria based on the proportion of missing data (less than 20% missing observations). Such variables include for example, information on pain [12, 13, 55], emotional functions and functions to handle stress [14, 15, 56], and skin functions [10, 11, 57], which have been shown to be highly prevalent and/or associated with functioning in

individuals with SCI and recovery after SCI. By including these variables in future network analyses, the relationships between variables may change and other influential patterns may be discovered.

## Study limitations

This study has several limitations. First, the SwiSCI Inception Cohort Study has been shown to be susceptible to potential non-response bias and item non-response bias: Fekete et al [17] have shown that women, older individuals and those with a non-traumatic SCI and a lower level of functioning are less likely to participate. Further selection bias may have occurred due to our specified inclusion criteria for the present study. Therefore, the generalizability of the results is limited. Second, for the trajectory analyses, missing observations were assumed to be missing at random–we assumed that missing observations in a person with SCI undergoing initial rehabilitation were either not related to functioning at all, or only related to the observed SCIM III measurements of that person (but not to the missing observation of SCIM III). Third, interpretation of identified functioning trajectory classes may be limited due to hetero-geneity within classes. In addition, we did not investigate potential predictors for class mem-bership due to the small sample sizes of identified functioning trajectories. This further limits the interpretability of functioning trajectories regarding important patient- and injury-related characteristics, such as age or injury severity, as well as their applicability in practice and corre-sponding studies are needed to clarify these questions. Moreover, such studies might in addi-tion investigate the relationship between trauma and acute care variables with functioning trajectory classes, as they have been shown to be associated with functioning recovery [58]. Fourth, the sample size of the identified functioning trajectory classes limited the subsequent network analysis to the *stable high* and the *moderate functioning improvement classes*. Thus, we were not able to draw conclusions regarding the comparison of networks across different func-tioning trajectory classes. In addition, we did not perform the planned multivariate trajectory analysis of the SCIM III subscales due to the results of the corresponding Rasch analyses. Thus, our knowledge on functioning trajectory classes with respect to specific SCIM III sub scores is limited. Fifth, the variable selection of relevant functioning problems was limited due to the number of missing observations in some variables related to the different consent sce-narios of the SwiSCI Inception Cohort. In addition, as described by Epskamp et al. [33] in the context of (psychological) network analysis, subsampling individuals based on a function of the observed variables (e.g., sum score) can lead to common effect structures in the network and to spurious unexpected negative edge weights, and thus, need to be interpreted with cau-tion [33]. Although we did not subset directly on the SCIM III sum score, but on the corre-sponding functioning trajectory classes, we may have introduced bias due to our variable selection procedure. Finally, the presented network analysis took an exploratory approach based on cross-sectional data. In particular, the reproducibility of our networks may be limited.

## Conclusion

This study has increased our knowledge about functioning trajectories of individuals with SCI undergoing initial rehabilitation in Switzerland. In addition to re-estimating and confirming previously identified trajectory classes, we also gained insight into the trajectory-specific preva-lence of relevant functional problems. Most importantly, by using a network analysis approach, we were able to represent and visualize the interconnectivity of functioning prob-lems at different time points during rehabilitation and identify potentially influential function-ing domains that could become trajectory-specific targets for clinical interventions. We believe

that this study is a first step towards a comprehensive picture of the complex developments during initial rehabilitation that can help health professionals understand the longitudinal course of functioning and how interventions on specific functioning areas may also improve abilities in other domains of functioning. Due to the exploratory nature of this study, further research is needed to confirm the findings presented. In a next step it will be important to study the role that patient- and injury-related characteristics play in this understanding. Ultimately, we hope that findings from the study may inform discussions about the application and use of functioning trajectories in clinical practice.

## Supporting information

**S1 Table. Overview about the selection of relevant functioning problems and corresponding SwiSCI variables according to Brief ICF Core Set for SCI (early post-acute situation) and selected categories according to Ballert et al.** Variables selected for the network analysis are marked in green.
(PDF)

**S2 Table. SCIM III items and scoring after Itzkovich et al, study labels and corresponding scoring system and ICF domains used in the study.**
(PDF)

**S3 Table. Expanded overview of available individuals who have consented to either the SwiSCI Inception Cohort or the use of routing clinical data for research purposes (MDS), as well as the excluded and included individuals for this study.**
(PDF)

**S4 Table. Main results of the Rasch analysis of the SCIM III total score.**
(PDF)

**S5 Table. Posterior classification table of the best-fitting latent process mixed model.**
(PDF)

**S6 Table. Estimated parameters of the best-fitting latent process mixed model.**
(PDF)

**S7 Table. Extended overview of study participant characteristics for the identified classes of functioning trajectories.**
(PDF)

**S8 Table. Edge weights of estimated mixed graphical model network for the moderate functioning improvement class.** A) T1. B) T4.
(PDF)

**S9 Table. Expected influence (z-scores) based on the mixed graphical model networks for the moderate improvement class.** A) T1. B) T4.
(PDF)

**S10 Table. Bridge expected influence (z-scores) based on the mixed graphical model networks for the moderate improvement class.** A) T1. B) T4.
(PDF)

**S1 Appendix. Summary of R Syntax for final trajectory and network models.** A) Final trajectory analysis model based on R package lcmm version 2.0.0. B) Final network analysis model for the moderate functioning improvement class based on R package bootnet version

1.5.
(PDF)

**S1 Fig. Observed individual functioning trajectories according to Rasch-transformed SICM III total scores (N = 1099).** Abbreviations: SCIM III, Spinal Cord Independence Measure version III.
(PDF)

**S2 Fig. Estimated parameterized link functions.** Abbreviations: AIC, Akaike information criterion; SCIM III, Spinal Cord Independence Measure version III.
(PDF)

**S3 Fig. Predicted mean functioning trajectories of latent process mixed models with fixed between-person trajectory variability across classes (set 1).** A) one class. B) two classes. C) three classes. D) four classes. E) five classes. F) six classes. Abbreviations: SCIM III, Spinal Cord Independence Measure version III.
(PDF)

**S4 Fig. Predicted mean functioning trajectories of latent process mixed models with varying between-person trajectory variability across classes (set 2).** A) one class. B) two classes. C) three classes. D) four classes. E) five classes. F) six classes. Abbreviations: SCIM III, Spinal Cord Independence Measure version III.
(PDF)

**S5 Fig. Estimated edge weight accuracy based on the mixed graphical model networks for the moderate functioning improvement class.** A) T1. B) T4. The lines on the y-axis indicate the different edges (names not shown) in the respective network and the x-axis indicated the corresponding edge weights. The red line shows the estimated edge weights based on the original sample of each network, the black dots and grey bars show the bootstrap mean edge weights and corresponding 95% bootstrap confidence intervals, respectively. Abbreviations: SwiSCI Swiss Spinal Cord Injury Cohort Study; T1, T4, SwiSCI assessment time points 1, 4.
(PDF)

**S6 Fig. Bootstrapped stability of centrality indices based on the mixed graphical model networks for the moderate functioning improvement class.** A) T1. B) T4. Abbreviations: SwiSCI Swiss Spinal Cord Injury Cohort Study; T1, T4, SwiSCI assessment time points 1, 4.
(PDF)

## Acknowledgments

We thank the SwiSCI Steering Committee with its members Xavier Jordan, Fabienne Reynard (Clinique Romande de Réadaptation, Sion); Michael Baumberger, Luca Jelmoni (Swiss Paraplegic Center, Nottwil); Armin Curt, Martin Schubert (Balgrist University Hospital, Zürich); Margret Hund-Georgiadis, NN (REHAB Basel, Basel); Laurent Prince (Swiss Paraplegic Association, Nottwil); Daniel Joggi (Representative of persons with SCI); Mirjana Bosnjakovic (Parahelp, Nottwil); Mirjam Brach, Gerold Stucki (Swiss Paraplegic Research, Nottwil); Carla Sabariego (SwiSCI Coordination Group at Swiss Paraplegic Research, Nottwil).

## Author Contributions

**Conceptualization:** Jsabel Hodel, Carla Sabariego, Cristina Ehrmann.

**Formal analysis:** Jsabel Hodel, Cristina Ehrmann.

**Methodology:** Jsabel Hodel, Carla Sabariego, Cristina Ehrmann.

**Supervision:** Carla Sabariego, Cristina Ehrmann.

**Visualization:** Jsabel Hodel, Cristina Ehrmann.

**Writing – original draft:** Jsabel Hodel.

**Writing – review & editing:** Jsabel Hodel, Carla Sabariego, Mayra Galvis Aparicio, Anke Scheel-Sailer, Vanessa Seijas, Cristina Ehrmann.

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
