## [Decision Letter · Decision Letter 0]

7 Nov 2023

PONE-D-23-29877Revisiting functioning trajectories in persons with spinal cord injury undergoing first rehabilitation in Switzerland: A network analysis approachPLOS ONE

Dear Dr. Hodel,

Thank you for submitting your manuscript to PLOS ONE. After careful consideration, we feel that it has merit but does not fully meet PLOS ONE’s publication criteria as it currently stands. Therefore, we invite you to submit a revised version of the manuscript that addresses all the points raised during the review process.  Please perform the required changes marked in red in the manuscript and give a point-to-point reply to all questions and comments of the reviewers. 

We look forward to receiving your revised manuscript.

Kind regards,

Antal Nógrádi, M.D., Ph.D., D.Sc.

Academic Editor

PLOS ONE

5. One of the noted authors is a group [SwiSCI Study Group]. In addition to naming the author group, please list the individual authors and affiliations within this group in the acknowledgments section of your manuscript. Please also indicate clearly a lead author for this group along with a contact email address.

Reviewers' comments:

Reviewer's Responses to Questions

**Comments to the Author**

1. Is the manuscript technically sound, and do the data support the conclusions?

Reviewer #1: Yes

Reviewer #2: Yes

2. Has the statistical analysis been performed appropriately and rigorously? 

Reviewer #1: Yes

Reviewer #2: Yes

3. Have the authors made all data underlying the findings in their manuscript fully available?

Reviewer #1: Yes

Reviewer #2: Yes

4. Is the manuscript presented in an intelligible fashion and written in standard English?

Reviewer #1: Yes

Reviewer #2: Yes

5. Review Comments to the Author

Reviewer #1: The title is OK but should be revised to highlight the association that is being studied between functional trajectories and the functioning domains. The following words “a network analysis approach” are not as important as the primary finding of this study. Especially, the network analysis is only performed on two groups of the four.

Overall impressions

The study aimed to evaluate functioning trajectories of individuals with SCI undergoing rehabilitation. The emphasis seems to be on categorizing individuals into one of four groups (high functioning, early, etc.). This study confirmed prior findings with updated datasets. While these findings are important, their contribution to improving clinical care is limited due to the lack of factors such as level and completeness of injury that were not considered in the functional recovery trajectory analyses. The study also identified functioning domains that affect the trajectory analysis. However, it is unclear if these functioning domains can be improved without considering the other SCI-related and patient-related factors towards functional improvement in these domains.

Introduction

This section is well written and highlights the article’s focus on tracking functional recovery and identifying functional domains related to recovery. While identifying functional domains associated with recovery is important, the article misses the opportunity to study SCI-related and patient-related characteristics that are associated with functional recovery.

Clinicians might not be able to predict if their treatment will yield better outcomes without the knowledge of how the functional recovery is affected by SCI-related (e.g. level and completeness) and patient-related characteristics (e.g. age).

Methods

The section clearly describes the time points and measures used in the study.

Rasch and Trajectory analyses are described clearly. Did the distribution of the functional recovery (within the dataset available to the researchers) affect the calibration.

Results

The Rasch analysis indicates the robustness of the SCIM III total scores for this population.

The trajectory analysis indicated the four-class model is preferred. Did the small sample size for early functioning and slow functioning affect the classification analysis. Would it have been better to do a two-class model for high functioning (including early function) and moderate to slow functioning.

Figure 3 does some analysis with sex and other variables such as injury severity. An analysis that includes both the injury severity (AIS) level and the neurological level of injury should be evaluated.

Discussion

Highlights their findings and the differences that were observed compared to previous research.

This section does a good job of comparing their findings with other research such as Tulsky et al. and the development of SCIM III.

The network analysis is interesting but is limited to just the two groups, which should be highlighted in this section as well.

As indicated in the limitation section a network analysis might connect sub-scores of SCIM III (domains) to the functional recovery trajectories but do not reflect the impact of specific rehabilitation interventions. This should be discussed in the article.

The section should highlight that a more detailed study is necessary to study the impact of other factors on functional recovery.

Conclusion

This section should indicate that while the prior results are confirmed their use in the clinical care is limited without considering the SCI-related and patient-related factors.

References:

While the article does a good job of citing various references. It misses the opportunity to highlight some key research that is associated with functional recovery in individuals with SCI in the last few years.

Early predictors of functional outcome after trauma. PM&R 2016;8(4):314–20.

Linking individual data from the spinal cord injury model systems center and local trauma registry: development and validation of probabilistic matching algorithm. Top Spinal Cord Inj Rehabil 2020;26(4): 221–31.

Evaluating associations between trauma-related characteristics and functional recovery in individuals with spinal cord injury. The Journal of Spinal Cord Medicine 2022.

Reviewer #2: A very extended documentation of functioning trajectories for SCI patients. Although, it remains in the era of one country and a specific Healthcare System, is is quite interesting.

The next step could be a comparative multi-country research to obtain a global aspect.

6. PLOS authors have the option to publish the peer review history of their article (what does this mean?). If published, this will include your full peer review and any attached files.

Reviewer #1: No

Reviewer #2: No

---

## [Author Response · Author response to Decision Letter 0]

15 Dec 2023

We thank the reviewers for their constructive and valuable feedback. Please find attached to this submission the PDF document "Response to Reviewers", which includes our response to each point raised by the reviewers and the journal requirements.

---

## [Editor Report · Decision Letter 1]

11 Jan 2024

Revisiting functioning recovery in persons with spinal cord injury undergoing first rehabilitation: Trajectory and network analysis of a Swiss cohort study

PONE-D-23-29877R1

Dear Dr. Hodel,

We’re pleased to inform you that your manuscript has been judged scientifically suitable for publication and will be formally accepted for publication once it meets all outstanding technical requirements.

Kind regards,

Antal Nógrádi, M.D., Ph.D., D.Sc.

Academic Editor

PLOS ONE

---

## [Editor Report · Acceptance letter]

1 Feb 2024

PONE-D-23-29877R1 

PLOS ONE

Dear Dr. Hodel, 

I'm pleased to inform you that your manuscript has been deemed suitable for publication in PLOS ONE. Congratulations! Your manuscript is now being handed over to our production team.

Kind regards, 

on behalf of

Prof. Antal Nógrádi 

Academic Editor

PLOS ONE